# Heterogeneity of the Immunological and Pathogenic Profiles in Patients Hospitalize Early Versus Late During an Acute Vital Illness as Shown in Native SARS-CoV-2 Infection

**DOI:** 10.3390/ijms26052349

**Published:** 2025-03-06

**Authors:** Krzysztof Laudanski, Ahmed Sayed Ahmed, Mohamed A. Mahmoud, Mohamed Antar, Hossam Gad

**Affiliations:** 1Department of Anesthesiology and Perioperative Care, Mayo Clinic, Rochester, MN 55905, USA; antar.mohamed@mayo.edu (M.A.); gad.hossam@mayo.edu (H.G.); 2Division of Pulmonary Care and Critical Care, Mayo Clinic, Rochester, MN 55905, USA; ahmed.ahmed5@mayo.edu (A.S.A.); mahmoud.mohamed@mayo.edu (M.A.M.)

**Keywords:** SARS-CoV-2, COVID-19, cluster analysis, immunological response, heterogeneity

## Abstract

The immune system’s response to an invading pathogen is the critical determinant in recovery from illness. Here, we hypothesize that the immune response will swiftly follow classical activation and a resolution trajectory in patients with the rapid evolution of symptoms if challenged by a viral pathogen for the first time. Alternatively, a dysregulated response will be signified by a protracted clinical trajectory. Consequently, we enrolled 106 patients during the first wave of COVID-19 and collected their blood within 24 h, 48 h, 7 days, and over 28 days from symptoms onset. The pathogenic burden was measured via serum levels of the S-spike protein and specific immunoglobulin titers against the S and N proteins of SARS-CoV-2. The nonspecific immunological response was gauged using interleukin 6, leukocytosis, and C-reactive protein. Coagulation status was assessed. Several serum biomarkers were used as surrogates of clinical outcomes. We identified four clusters depending on the onset of symptoms (immediate [A], 6 days [B], 12 days [C], and over 21 days [D]). High variability in the S-spike protein in cluster A was present. The corresponding immunoglobulin titer was random. Only procalcitonin differentiated clusters in terms of markers of nonspecific inflammation. Coagulation markers were not significantly different between clusters. Serum surrogates on cardiomyopathy and neuronal pathology exhibited significant variability. Implementation of ECMO or noninvasive ventilation was more prominent in cluster C and D. Interestingly, SOFA or APACHE II scores were not different between nominal (A and B) versus dysregulated clusters (C and D).

## 1. Introduction

When faced with a pathogen, the immune system can react in multiple ways [1,2,3,4,5]. The ideal response eliminates the pathogen and minimizes collateral damage. Consequently, symptoms can range from asymptomatic to severe or even fatal, reflecting the adequacy of immune activation [6,7]. Initially, a viral challenge triggers nonspecific immune responses, which depend on the pathogen’s load and virulence interacting with the host’s immune makeup. Common indicators of this early response include leukocytosis, C-reactive protein (CRP), and interleukin-6 (IL-6) providing rapid defense against viral intrusion [8,9]. An effective deactivation of this blunt, nonspecific immunity relies on the emergence of specific immunity indicated by the production of specific antibodies (starting with IgM and IgA, followed by IgG) [1]. However, if these initial responses are not well regulated, collateral damage may emerge [10,11,12]. Excessive complement utilization, inadequate coagulation cascade activation, and a dysregulated immune response demonstrate loss of homeostasis and an increase in surrogate markers of organ damage or outright organ dysfunction [3,12,13,14].

The COVID-19 pandemic provided a unique opportunity to study immune responses without prior immunological memory, and thus reducing the heterogeneity of immunological response due to prior exposure. By focusing on the onset of signs of severe immunological responses in symptomatic COVID-19 patients, we hypothesized interindividual variability in the onset of symptoms based on native immunological response severity untainted by immunological memory [1,3,10]. Specifically, we anticipated that patients with extended symptom duration might represent a subgroup where the immune system either failed to control the infection due to an ineffective response or overreacted, causing collateral damage.

To investigate these hypotheses, we focus on patients experiencing SARS-CoV-2 infection during the first wave of the pandemic in 2019, before vaccinations were made available [6,7]. We speculated that individuals with extended, unresolved symptoms and requiring late hospitalization may represent cases of dysregulated or ineffective immune responses. Such inadequate response will be presented by either inadequate viral suppression (elevated levels of S-protein), leading to organ damage (HMGB-1, RAGE, TnI, neuro-injury and neuro-degeneration markers, cardiac injury surrogates), or by an overactive non-specific immune response (IL-6, procalcitonin, CRP) that fails to establish targeted immunity (measured as level of specific immunoglobulins) [3,10,11].

## 2. Results

### 2.1. Time-Cluster of Individuals Depending on the Time of Hospitalization

Unsupervised cluster analysis using *k*-*means* was conducted to determine patient groups with different relationships between the emergence of self-noticed symptoms and hospitalization. Initially, we characterized five time-centered clusters when the onset of the symptoms was analyzed (F [4;101] = 795.888; *p* < 0.0001). Cluster A included patients (n = 41) when the onset of the symptoms and admission had almost no time separation (Figure 1). Cluster B was characterized by patients (n = 41) with an average delay of six days between the onset of the symptoms and admission to the hospital. Cluster C (n = 13) aggregated patients with the average onset of symptoms being 12 days. Finally, we merged two clusters (cluster D1 = 22 days; n = 4 and cluster D2 = 32 days; n = 7) into one (cluster D) with an average onset of symptoms over 21 days or more, considering the minimal separation of the cluster groups.

### 2.2. Characterization of Demographical and Clinical Variables Between Clusters

The demographical and baseline clinical characteristics of the individuals in each cluster are visualized in Table 1. Over-representation of males was seen in clusters A, C, and D, while cluster B was characterized by an even split of both genders (χ^2^ [106;6] = 14.38; *p* = 0.026). Race did not differentiate clusters (χ^2^ [106;6] = 11.93; *p* = 0.063). Age did not correlate with t_adm_, and the frequency of individuals over age 60 differed between clusters (χ^2^ [106;6] = 2.58; *p* = ns). There was no difference in BMI at t_adm_ between clusters.

CCI was not different between clusters (F [3;105] = 7.12; *p* = ns). Cluster A had more cases of peripheral vascular disease (χ^2^ [106;3] = 9.015; *p* = 0.029). Cluster D had the few individuals with leukemia, which, coupled with the few cases in this cluster overall (n = 11), rendered this finding significant (χ^2^ [106;6] = 8.719; *p* = 0.033) (Table 1).

### 2.3. Infectious Patterns for Cluster Patients

Cluster B demonstrated the highest serum S-spike protein level at admission compared to other clusters (Figure 2A). The level of S-protein remained variable in the follow-up time intervals, with some upward trajectory in group D (Figure 2B).

We found a significant difference in the admission IgA level against S and N protein (KW [3;39] = 9.26; *p* = 0.026) in clusters A and B compared to other clusters 7 days after admission (Figure 3A). Cluster A had significantly diminished levels of IgA, while cluster D had increased type A immunoglobulins (Figure 3A). The follow-up trajectory of the immunoglobulin types demonstrated (Figure 3A) significant longitudinal variability. Titer of IgM shows minimal variability (Figure 3B). There was no difference at the admission of IgG, but significant differences were noted between clusters (Figure 3C). However, during hospital IgG, there was considerable variability in cluster B (Figure 3C).

### 2.4. Immune and Coagulation Features of Each of the Clusters

Serum levels of alarmin (HMGB-1, RAGE) did not differ between time points and clusters themselves (Appendix A). Regarding nonspecific immunological responses, leukocytosis, CRP, or serum levels of IL-6 or procalcitonin at admission were not a differentiating factor (Figure 4). The trajectory of the markers over time demonstrated significant variability in procalcitonin, which was observed only within cluster B (Figure 4).

Coagulation parameters revealed no difference between clusters at both t_adm_ and subsequent time points (Appendix A). There is non-significant variability across the coagulation variables within each cluster over time.

### 2.5. Symptomology Presentation of Patients in Different Clusters

Cough, fever, and breathing difficulty were the most prevalent symptoms across all clusters (Figure 5). Clusters B, C, and D had the highest frequency of the symptoms as compared to cluster A. Cluster A had a moderate expression of symptoms, with cough, fever, and breathing problems being the most common, while fatigue and muscle pain were the next most common but not very prevalent. Cluster B demonstrated chills, fatigue, muscle and chest pain, diarrhea, nausea and vomiting. The emergence of chest pain and loss of smell characterized cluster C. Finally, cluster D showed markedly increased confusion and diarrhea.

### 2.6. Biological Markers of End-Organ Damage Across Clusters

Markers of cardiac strain (NT-BNP; TnI) demonstrated significant variability over the time between time points (Figure 6A,B). Cluster C showed increased cardiac damage markers at the third and fourth blood draw, while cluster D showed the biphasic evolution of the levels.

Neurodegeneration marker of tau also demonstrated significant time variability (Figure 7C). When looking at the markers of neurological injury, we found no changes in amyloid β level or tau over time or between the clusters (data not shown). However, we did find elevation in NCAM-1 at acute and 48 h after hospitalization, indicating peripheral nerve injury. Neurogranin demonstrated similar characteristics (Figure 7).

### 2.7. Clinical Outcomes Across Clusters

ECMO was more prevalent in clusters C and D (χ^2^ [3;106] = 11.205; *p* = 0.011) (Table 1). Noninvasive ventilation is more prevalent in (χ^2^ [106;9] = 20.109; *p* = 0.017) (Table 1). There was a tendency for a statistically significant difference in overall length of stay (K [106] = 7.196; *p* = 0.066) and length of mechanical ventilatory support (K [103] = 6.316; *p* = 0.097) and duration of ECMO (K [106] = 11.054; *p* = 0.011), particularly when A vs. D clusters were compared (Figure 1B). Cluster D represented patients with higher APACHE_24hr_ II admission scores (post hoc cluster A vs. D; *p* = 0.01) (Table 1).

SOFA or MODS scores did not differ between clusters during the first five days.

The clinical trajectories of these patients demonstrated no increased prevalence of cerebrovascular events (χ^2^ [106;9] = 5.019; p = ns), deep venous thrombosis (χ^2^ [106;3] = 1.269; *p* = ns), and pulmonary embolism (χ^2^ [105;3] = 6.259; *p* = ns) at one and six months after discharge. Mortality at one month was highest in clusters C and D (χ^2^ [104;3] = 9.937; *p* = 0.019), but not at six months (χ^2^ [96;3] = 6.029; *p* = ns). Disposition at one month was no different between the studied clusters (χ^2^ [101;9] = 12.989; p = ns).

## 3. Discussion

In this cohort observational study of individuals infected with SARS-CoV-2 infection during the first wave of the pandemic, we defined dysregulated immune responses based on symptom duration and clinical presentation rather than solely on cytokine or serum biomarker profiles. This sets us apart from other researchers that emphasize biochemical markers alone [9,11,15]. Conversely, patients who presented at the hospital early after symptom onset and experienced rapid recovery were examples of effectively regulated immune responses. These patients’ immune systems could mount an effective antiviral response without prolonged inflammatory consequences [3,4,5,11]. This distinction may be crucial for understanding how immune system timing and efficacy relate to overall outcomes in COVID-19 patients, emphasizing the role of early, targeted immunity in favorable clinical trajectories.

Our research showed patients who experience illness-related symptoms, forcing them to check in the hospital, in four distinct time groups. Cluster A represented patients with an immediate onset of symptoms that necessitated medical assistance. These patients were relatively quickly discharged from the hospital. Cluster B included patients hospitalized after a delay following the onset of severe symptoms. Clusters C and D represented patients with significantly delayed hospitalization, thus extended course of illness. These clusters represent the three different naïve and native responses to viral infection [2,6,16]. Our study assumes that an effective immune response triggers symptoms shortly after exposure [1,2,11]. Conversely, delayed hospitalization indicates a failure to recover from a viral infection, likely due to an unfavorable immune response. An increased level of viral protein load indicates increased toxicity and pathogenic load [17,18]. Rising immunoglobulin titers show an acquisition of adaptive immunity [15]. However, when analyzing viral burden, only cluster C showed an increased S-protein burden at admission, while IgA, IgM, and IgG levels against SARS-CoV-2-related antigens did not differ significantly between patients. The serum post-infection evolution of CRP and IL-6 levels showed no significant variation between clusters, which is consistent with the nonspecific response to viral infection or immune challenges in general [1,2,16,19]. The most important differences were in serum surrogates of organ damage and clinical trajectory. We found notable differences in serum markers of cardiac overload and injury (NT-BNP, TnI), atypical markers of central nervous system damage (NRNG), and biomarkers of peripheral nerve injury (NCAM-1) [16,20]. The release of these markers was more heterogeneous compared to innate and acquired immunity dynamics and markers of nonspecific tissue injury (HMGB-1, RAGE). These data are consistent with prior data that end organ damage and related biomarkers are frequently elevated in severe illness.

The data show that using unsupervised clustering to classify patients admitted to the hospital reveals similar immunological responses but significant differences in clinical symptoms and biomarkers of end-organ dysfunction and injury. This indicates that individual predisposition and organ susceptibility play a more crucial role in patient outcomes than the immune response itself. We saw profound immune reactions, but their heterogeneity is lower than expected [10,18]. This interpretation aligns more with the danger theory of immune response rather than the conventional belief that a dysregulated immune response causes organ damage and dysfunction [2,21]. Dysregulation theory suggests variability in immune responses yet does not define a “normative” response through quantifiable lab values. Consequently, we should change our thinking from dysregulation in response to dysregulated organ damage [4,5,6,16].

Our study has some unique aspects. It examines the response of an immunologically naïve population to a novel pathogen [16]. Unlike other common infections like influenza or streptococcus, SARS-CoV-2 had little prior exposure or vaccine priming in the population. The entire enrollment took place during the initial wave of SARS-CoV-2 infection when vaccines were not available, and one strain of the virus was predominant [22]. This allowed us to observe immune responses without prior memory of the viral challenge. We used an unsupervised cluster analysis to classify patients based on the delay between symptom onset and hospitalization. Although this method has its limitations, it avoids many common biases by being agnostic in approach [23,24]. We did not use a specific cut-off for immune system activation, such as IL-6 levels, which are highly variable and often fail to correlate with symptoms or signs of organ damage [11,13,14]. Instead of selective marker analysis, our cluster analysis allowed us to integrate several markers. This approach aligns well with viewing the immune system as a multidimensional system consisting of interdependent parts [1,2,3,14,16,17,19]. However, our ability to predict responses and dynamic intermediary elements remains limited, preventing us from demonstrating a theoretical immune system response in its heterogeneity.

Our study has several limitations. Even though the recruitment period was short, patients could be exposed to other SARS-CoV-2 strains or acquire another viral infection. We do not have sequencing data to confirm specific strains being acquired. We surveyed people only if they remained sick enough to be in the hospital or had no symptoms but checked into the hospital (cluster A). Only motivated or very ill patients were followed up, leading to a bias towards sicker individuals. Naturally, less illness-stricken or deceased patients we dropped from the study. Although we contacted them by phone after recruitment, the response rate was low. Conversely, patients in cluster A had fewer symptoms, yet their immune markers were like other groups. Another bias is racial bias, and a lack of insight into social determinants of health [6]. Our patient population reflects a typical large American city with a disproportionately high number of African American patients. This group has a high prevalence of cardiac illness, diabetes, smoking, and inadequate social support or housing. Given the even distribution of these patients across all clusters, the effects of these determinants on hospitalization and inflammatory response were minimal. We also experienced significant dropouts in blood draws after six months, which was expected due to the challenging demographics of our population. Additionally, we could not confirm the disposition of our patients at six months due to difficulties establishing contact, which could be due to study withdrawal or other reasons. Finally, SARS-CoV-2 infection may be unique, limiting the generalizability of our findings.

Our study contains inherent assumptions. We equated symptoms with the severity of illness, but these symptoms were several enough to drive patients to seek medical attention. However, whether the patient elected to come to hospital depends on illness severity and the patient’s ability to cope with the burden of illness. We did not measure each patient’s resiliency, but it is reasonable to assume people exposed to the pandemic would respond similarly in terms of awareness of their symptoms. Symptoms may not always correlate with the immune response or infection profile, which is often confirmed by research. As our study demonstrates, the lack of correlation between a host’s response to a viral challenge and symptomatology should not be surprising. Additionally, symptoms are influenced by various factors, which are impossible to collect in their entirety.

## 4. Materials and Methods

### 4.1. Recruitment of Individuals for the Study

Institutional Review Boards (IRBs; #813913) of the University of Pennsylvania approved the study. Patients were approached for consent within 24 h after admission if they had positive SARS-CoV-2 testing. We enrolled 106 patients between April 2019 and September 2019, and the observation period was finished before December 2019 (Table 1).

### 4.2. Symptoms Data

Each patient was provided with a link to the RedCAP™ survey to gauge the onset and nature of the symptoms. They were asked to relate the intensity and duration of the symptoms before hospitalization. Each patient was offered sufficient time to fill out the survey.

### 4.3. Clinical Data

The electronic medical records (EMRs) were extracted for demographic and medical data. Because of the very low prevalence of certain individuals, we classify participants into Caucasian, Black, and Other/Asian/Unknown. The burden of chronic disease was calculated using the Charlson Comorbidity Index (CCI) [25].

The Acute Physiology and Chronic Health Evaluation II (APACHE II) was calculated within one hour (APACHE_1hr_) and at 24 h after admission (APACHE_24hrs_) [26]. The severity of the illness was determined via the Sequential Organ Failure Assessment (SOFA) [26]. Survival was determined at 28 days and 6 months from admission. The incidence and nature of cerebrovascular accident (CVA), deep venous thrombosis (DVT), and pulmonary embolism (PE) were determined from medical records. Renal failure was calculated using RIFLE criteria. The treatment with remdesivir, convalescent plasma, and steroids was determined from electronic health records (EHRs). These treatments were highly protocolized as per hospital policy and according to the FDA recommendations for the given treatment. Leukocytosis was calculated by extracting white blood cell (WBC), platelet count, d-dimers, international normalized ratio (INR), procalcitonin (PCT), and C-reactive protein (CRP) from medical records.

### 4.4. Sample Collection and Processing

Each patient had a blood draw performed at predetermined time points, including within 48 h after admission (t_baseline_), followed by another 48 h interval (t_48hr_), then 7 days (t_7d_), and finally at 3-month follow-up (t_3m_). The first three samples were collected if the patient remained in the hospital. The last sample was collected in inpatient and outpatient settings and was dependent on patient availability.

Blood was collected through venipuncture or a central access line into a BD Vacutainer™ heparinized collection tube (BD, Brunswick, NJ, USA). Blood was spun at 2000× *g* at 4 °C for 10 min to isolate serum. The obtained serum was then aliquoted and stored at −80 °C until biochemical assays after inactivation with the addition of Triton X-100.

### 4.5. Assessment of Pathogen Burden and Specific Humoral Response

The level of S-protein was measured using commercially available kits (Ray Biotech, Stanford, CA, USA). The level of specific immunoglobulins against proteins S and N were measured using a commercially available absorption assay and compared to nonspecific binding against bovine serum albumin (Ray Biotech, Stanford, CA, USA). Notably, none of the patients was vaccinated against SARS-CoV-2, considering the timing of collection.

### 4.6. Assessment of Nonspecific Immune and Coagulation Response

Serum IL-6 was measured using commercial kits (Biolegend, San Diego, CA, USA). The absorbance OD values at 450 nm, with reference values at 570 nm, were subtracted and referenced against the standard curve. TnI, NT-BNP, BCL-2, neurogranin, and NCAM-1 were assessed with Luminex technology (Luminex, San Diego, CA, USA).

### 4.7. Statistical Analysis

The Shapiro–Wilk W test and distribution plots were used to test the normality of distribution variables. The obtained lab values were transformed to the z-scores using an entire population of raw scores for a given variable. Parametric variables will be expressed as mean ± SD (X ± SD) and compared using *t*-Student [n]. Unsupervised cluster analysis was conducted using k-means to assess the distance of similarity. For non-parametric variables, median (M_e_) and interquartile ranges (IR) will be shown, and the U–Mann–Whitney statistic (U [df;n]) will be employed to compare such variables. ANOVA was calculated for parametric variables with multiple discrete values with post hoc tests to assess individual contrasts. Kruskal–Willis (KW [df;n]) test was used for similar data in case of their non-parametric nature. Paired contrasts were estimated when applicable with listwise data removal if missing information was uncovered. Correlational momentum was calculated as *r^2^* Pearson value. The regression analysis was conducted through stepwise methods. Clusters were generated using an unsupervised *k*-means algorithm.

Paired contrasts were estimated when applicable with listwise data removal if missing information was uncovered. Statistically significant difference was considered as two-tailed *p* < 0.05 unless a specific one-sided hypothesize was formed in the text

Statistical analyses will be performed with SPSS 29 (IBM, Waltham, NY, USA).

## 5. Conclusions

In conclusion, our study showed that while different patient clusters experienced worsening COVID-19 symptoms leading to hospitalization, the immunological and viral dynamics remained consistent across groups. The primary variations were in symptoms, end-organ dysfunction and injury biomarkers, and clinical trajectories. This indicates that an individual’s organ response is the key factor rather than the progression of the immune response itself.

## Figures and Tables

**Figure 1 ijms-26-02349-f001:**
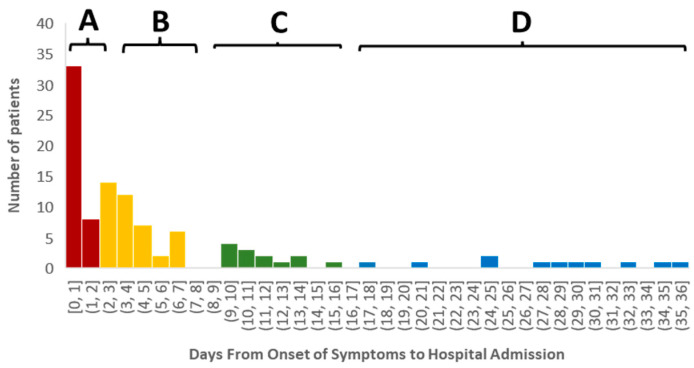
Patients clustered in four groups depending on the onset of the symptoms to hospitalization. Cluster A included patients (n = 41) when the onset of the symptoms and admission had almost no time separation (tint = 0) (Figure 1). Cluster B was characterized by patients (n = 41) with an average delay of six days between the onset of the symptoms and admission to the hospital. Cluster C (n = 13) aggregated patients with the average onset of symptoms being 12 days. Cluster D showed patients with an average onset of symptoms over 21 days or more (n = 11).

**Figure 2 ijms-26-02349-f002:**
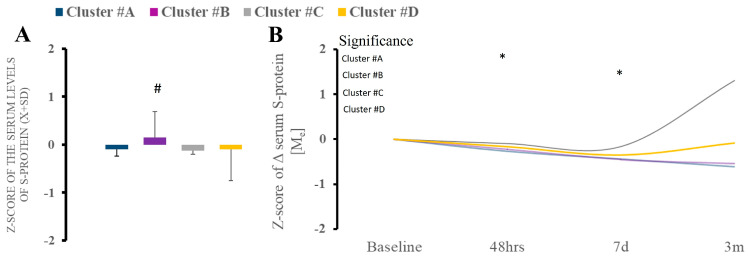
(**A**,**B**). There were differences in serum S-spike protein between clusters at baseline, with a significant increase in cluster B compared to other clusters. * Statistically significant differences between baseline and subsequent time points within the cluster at a given sampling time point.

**Figure 3 ijms-26-02349-f003:**
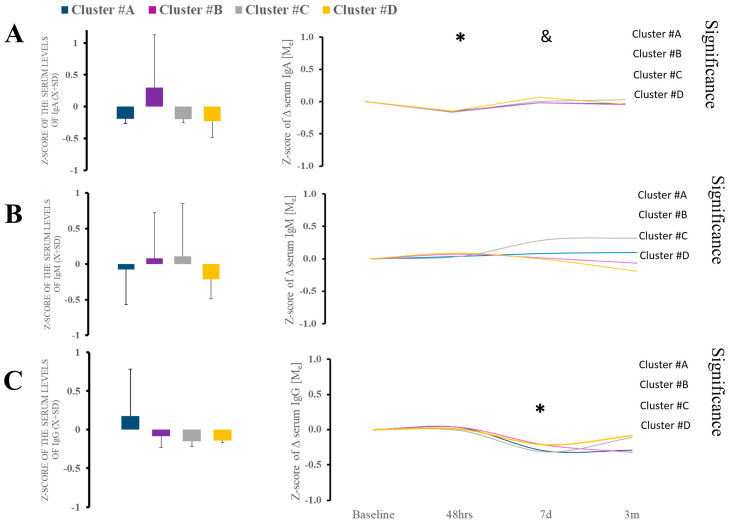
(**A**–**C**). There were no differences in serum IgA, IgM, and IgG between clusters at baseline. Analysis of the IgA and IgG revealed some variability, but it was minimal. * Statistically significant differences between baseline and subsequent time points within the cluster at a given sampling time point. & Statistically significant differences between clusters at a given sampling time point.

**Figure 4 ijms-26-02349-f004:**
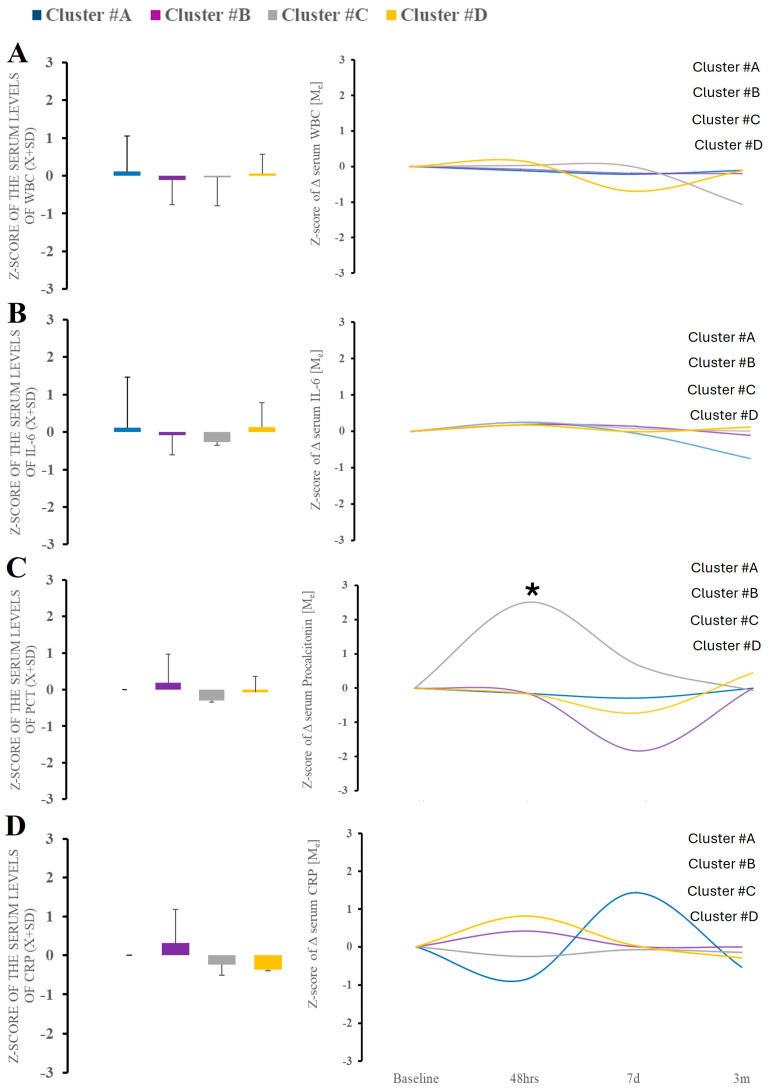
There were no differences in serum WBC, IL-6, and CRP between clusters at baseline (**A**–**D**). Analysis of the PCT demonstrated significant variability in cluster B. * Statistically significant differences between baseline and subsequent time points within the cluster at a given sampling time point.

**Figure 5 ijms-26-02349-f005:**
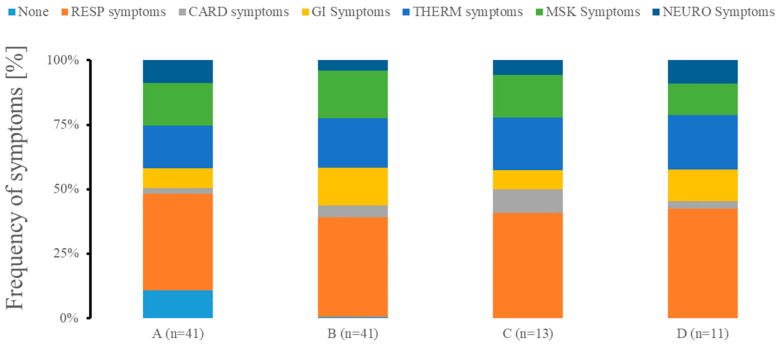
Symptomology presentation of patients in different clusters at admission. We grouped patients with none symptoms (NONE) as well as exhibiting predominantly respiratory problems (RESP; cough, runny nose, and breathing difficulty), cardiac system-related symptoms (CARD; chest pain), gastrointestinal symptomatology problems (GI; diarrhea, nausea, and vomiting), abnormalities in temperature management (TEMP; fever and chills), predominant musculoskeletal issues (MSK; fatigue and muscle pain), or neurological issues (NEURO; headache, loss of smell, and confusion). n represents the number of patients in each of the clusters.

**Figure 6 ijms-26-02349-f006:**
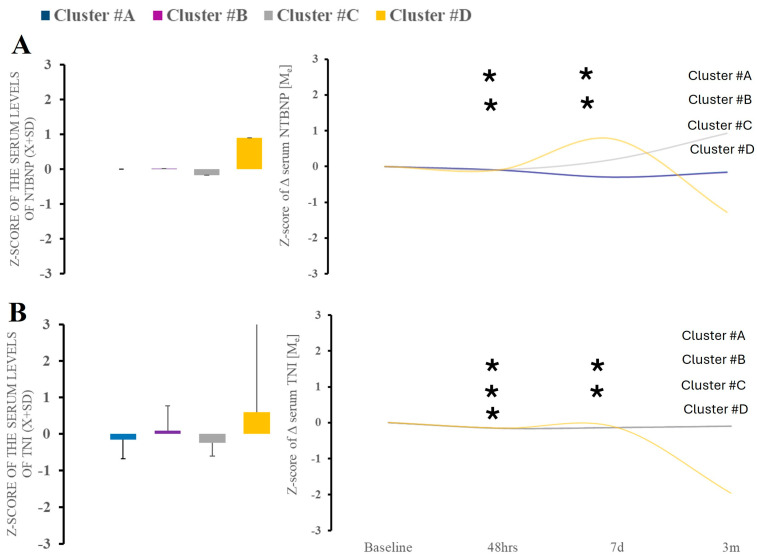
There were no differences in serum NT-BNP or TnI between clusters at baseline (**A**). Time-wise analysis within markers revealed several differences across all studied clusters except (**B**). * Statistically significant differences between baseline and subsequent time points within the cluster at a given sampling time point.

**Figure 7 ijms-26-02349-f007:**
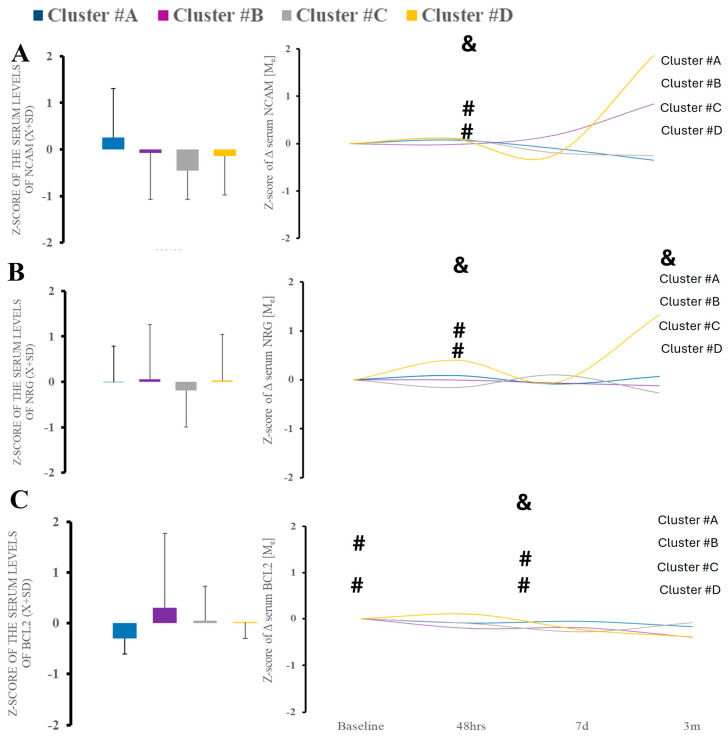
There were no differences in serum surrogates of peripheral nerve injury (NCAM-1) (**A**), and neurodegeneration (NRNG, BCL-2) (**B**,**C**). Time-wise analysis within markers revealed several differences across all studied clusters except B. &—statistical difference when comparing clusters at the same time point; #—statistical difference when comparing to cluster A.

**Table 1 ijms-26-02349-t001:** Demographical and clinical characteristics of the study sample in general and patients in each symptomatic cluster.

	Total (n = 106)	Cluster A (n = 41)	Cluster B (n = 41)	Cluster C (n = 13)	Cluster D (n = 11)	Sig (*p*-Value)
Age [X ± SD]	58.95 ± 18.23	55.2 ± 19.8	62.5 ± 14.6	58.46 ± 16.9	60.3 ± 24.2	0.56
Over 60 (%)	56.6%	56.10%	58.50%	46.20%	63.60%	0.86
BMI [X ± S.D.]	31.71 ± 8.91	30.5 ± 6.8	32.7 ± 9.4	34.5 ± 12.6	28.5 ± 7.8	0.51
Gender						
Male (%)	59.5%	70.70%	46.30%	69.20%	54.50%	
Female (%)	39.6%	29.30%	53.70%	30.80%	36.40%	0.026
Not Reported (%)	0.9%	0%	0%	0%	9.10%	
Race						
White/Caucasian/Hispanic Latino [%]	27.3%	29.20%	19.50%	53.90%	18%	
Black [%]	62.3%	65.90%	70.70%	30.80%	54.50%	0.171
Other/Asian/unknown [%]	10.3%	4.90%	9.70%	15.40%	27.30%	
**Clinical trajectory**						
Admitted to the I.C.U. [%]	50%	43.90%	51.20%	53.80%	63.60%	0.68
Noninvasive Intubated [%]	33%	26.80%	31.70%	38.50%	54.50%	0.36
Extracorporeal membrane oxygenation [%]	9.4%	0%	9.80%	23.10% *	27.30% *	0.011
Length of stay in the Hospital [X ± S.D.]	17.51 ± 26.122	11.4 ± 13.6	16.3 ± 24.7	28.8 ± 46	31.6 ± 29.9	0.066
Length of stay in the I.C.U. [X ± S.D.]	10.92 ± 24.533	5.63 ± 12.6	8.78 ± 19.6	27.91 ± 48.5	23.44 ± 34	0.182
APACHE SCORE 1 h [X ± S.D.]	11.13 ± 7.809	9.6 ± 7.1	9.1 ± 8.3	10.23 ± 5.4	15.82 ± 9.7	0.22
APAHE SCORE 24 h [X ± SD]	11.04 ± 7.313	10.1 ± 8.2	10.6 ± 6.3	11.2 ± 6.5	15.6 ± 7.7	0.079
**Preexisting conditions**						
Myocardial infarction [%]	5.7%	9.80%	4.90%	0%	0%	0.428
Congestive heart failure [%]	15.1%	9.80%	14.60%	15.40%	36.40%	0.187
Peripheral vascular disease [%]	7.5%	9.80%	2.40%	0%	27.30%	0.029
Cerebrovascular stroke [%]	11.3%	7.30%	14.60%	7.70%	18.2	0.617
Chronic obstructive pulmonary disease [%]	14.2%	17.10%	14.60%	7.70%	9.10%	0.8
Diabetes mellitus [%]	34.9%	31.70%	39.00%	23.10%	45.50%	0.61
Chronic kidney disease [%]	24.5%	22.00%	29.30%	23.10%	18.20%	0.82
Solid tumor [%]	10.4%	9.80%	14.60%	7.70%	0.00%	0.53
**Smoking status**						
Smoker [%]	9.43%	17.10%	4.90%	0.00%	9.10%	
Former smoker [%]	32.08%	24.40%	39.00%	38.50%	27.30%	0.38
Non-smoker [%]	58.49%	58.50%	56.10%	61.50%	63.60%	

* showing statistical difference in post-hoc analysis between groups.

## Data Availability

Data are available upon request and after appropriate approval by the IRB.

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
