# Peer review of "Heterogeneity of the Immunological and Pathogenic Profiles in Patients Hospitalize Early Versus Late During an Acute Vital Illness as Shown in Native SARS-CoV-2 Infection"

_ijms, 2025, doi:10.3390/ijms26052349_

Round 1
Reviewer 1 Report
Comments and Suggestions for Authors
This is an interesting paper describing the immune system of patients naïve to the Sars-Cov -2 virus.
The paper, however, needs major revision:
Title - if researchers only studied patients with COVID-19 infection please put this in the title
Abstract - please divide into distinct sections - background, materials and methods, results, and conclusion.
Introduction: please write the purpose of the paper at the end
The text from line 60 to 69 is a de facto description of the results and conclusions of the study and should be moved to the discussion chapter.
subsection materials and methods please move before the results section.
Discussion - Please add the name of the tested virus in the first sentence of the discussion.
English needs minor revision to make the content more accessible to the reader
Author Response
- This is an interesting paper describing the immune system of patients naïve to the Sars-Cov -2 virus.
- We appreciate your feedback and look forward to improving our work.
- Title - if researchers only studied patients with COVID-19 infection, please put this in the title
- This work uses COVID-19 to show how infection profiles change over time. Adding this to the title may distract from the manuscript's main point. We have added sentences throughout to emphasize this. If the reviewer insists, we will reconsider changing the title.
- Abstract - please divide into distinct sections - background, materials and methods, results, and conclusion.
- This is a requirement of the journal. We will be happy to adjust it if the editorial office permits us to do so.
- Introduction: please write the purpose of the paper at the end. The text from line 60 to 69 is a de facto description of the results and conclusions of the study and should be moved to the discussion chapter.
- This is a good point. We moved the section to the discussion and made the introduction more concise.
- For the subsection materials and methods, please move before the results section.
- The journal editorial policy forced us into the presented arrangement.
- Discussion - Please add the name of the tested virus in the first sentence of the discussion.
- Adjusted as requested by the reviewer. We also rewrote a significant part of the discussion.
Reviewer 2 Report
Comments and Suggestions for Authors
In the present work, the authors describe the immune response over time in patients infected with SARS-CoV-2, categorizing them into clusters based on the time elapsed between symptom onset and hospitalization. The article is suitable for the "Molecular Immunology" section to which it was submitted and could be of great interest to many readers. However, some key information is missing from the article, which loses significance without these details. Therefore, a major revision is essential before considering the manuscript for publication.
Major revisions:
When was the study conducted? In 2021, an intense vaccination campaign against SARS-CoV-2 began, so from that period onward, many patients would have pre-existing immunity. The authors should specify the time period during which the patients were recruited, whether they were vaccinated, with which vaccine, how many doses they received, and how much time had passed since the last dose. Additionally, the authors should report whether the patients had previously contracted a natural infection.
Given the differences in the patients' symptoms, it is crucial to know whether all the patients were infected with the same strain of SARS-CoV-2. In the absence of viral sequencing data, the period during which the patients were recruited might provide important clues about the dominant viral variant. This information should be included in the manuscript.
The authors quantified several parameters in the serum of the patients (e.g., IL-6, S-protein, S and N immunoglobulins, BCL-2). However, the manuscript only reports the Z-score values in the figures. To add significance to the manuscript, the authors should include a figure for each measured parameter, showing a graph with the values for each sample analyzed, appropriately divided into the different clusters. This is absolutely crucial for understanding the manuscript. The Z-score graphs can be included but should be redrawn to make them more comprehensible. In fact, the graphs are not understandable for people who cannot distinguish colors, and the statistical representation is very complex (it is unclear which samples the different *, &, # refer to). For better comprehension of the graphs, it is also recommended to include the horizontal axis.
Minor revisions:
Title: The title of the manuscript refers to a generic "viral infection," but the data presented only concern SARS-CoV-2-positive patients. This should be specified in the title.
Results 2.1, lines 78-83: The authors should clarify whether the patients in clusters C and D required hospitalization.
Figure 5: Which time points does the figure refer to?
MeM 4.4: How was the virus in the blood of infected patients inactivated?
MeM 4.6: Which commercial kit was used to measure IL-6?
MeM 4.7: It would be useful to specify how the Z-score and the Z-score of Ϫ were calculated.
Author Response
We appreciate the comments provided by this reviewer. Many of them aligned with our considerations during the initial writing of the manuscript. We have reviewed the entire manuscript and enhanced several points based on the recommendations given.
- When was the study conducted? In 2021, an intense vaccination campaign against SARS-CoV-2 began, so from that period onward, many patients would have pre-existing immunity. The authors should specify the time period during which the patients were recruited, whether they were vaccinated, with which vaccine, how many doses they received, and how much time had passed since the last dose. Additionally, the authors should report whether the patients had previously contracted a natural infection.
- This is a crucial point that we should emphasize more prominently throughout the paper. All patients were recruited between April 2020 and September 2020, during which no vaccine was available. The vaccine became available in December 2020 and was initially restricted to medical personnel. Given the short recruitment period, the risk of coinfection was low. We have incorporated this point into the introduction, materials, methods, results, and discussion sections.
- Given the differences in the patients' symptoms, it is crucial to know whether all the patients were infected with the same strain of SARS-CoV-2. In the absence of viral sequencing data, the period during which the patients were recruited might provide important clues about the dominant viral variant. This information should be included in the manuscript.
- The recruitment period was short. As a result, there is a possibility that all patients were infected with the same strain. Although we do not have sequence data, we will address this comment in the discussion section.
- The authors quantified several parameters in the serum of the patients (e.g., IL-6, S-protein, S and N immunoglobulins, BCL-2). However, the manuscript only reports the Z-score values in the figures. To add significance to the manuscript, the authors should include a figure for each measured parameter, showing a graph with the values for each sample analyzed, appropriately divided into the different clusters. This is absolutely crucial for understanding the manuscript. The Z-score graphs can be included but should be redrawn to make them more comprehensible. In fact, the graphs are not understandable for people who cannot distinguish colors, and the statistical representation is very complex (it is unclear which samples the different *, &, # refer to). For better comprehension of the graphs, it is also recommended to include the horizontal axis.
- We spent 6 months on this manuscript to optimally present complex data. Using direct data graphs, as suggested, was challenging due to varying analyte ranges (pg/ml, ng/ml, mcg/ml) and their dynamics, resulting in multiple axes making the graph hard to read. Z-scores standardized all values, simplifying comparison. We believe this approach effectively shows the evolution of viral response in patients experiencing their first immune challenge. In the process of submitting this review, we consulted a statistician and graphic designer, but after several iterations, we found the current presentation most clear yet still not easy to read.
- Graphs were enhanced for individuals with color blindness by adding red and green patterns while retaining yellow and blue.
- Horizontal axes were not added as they impeded visual recognition or were too difficult to see properly.
- Title: The title of the manuscript refers to a generic "viral infection," but the data presented only concern SARS-CoV-2-positive patients. This should be specified in the title.
- This work uses COVID-19 to show how infection profiles change over time. Adding this to the title may distract from the manuscript's main point. We have added sentences throughout to emphasize this. If the reviewer insists, we will reconsider changing the title.
- Results 2.1, lines 78-83: The authors should clarify whether the patients in clusters C and D required hospitalization.
- We put the comment
- Figure 5: Which time points does the figure refer to?
- This was clarified with statement that there were admission symptoms
- MeM 4.4: How was the virus in the blood of infected patients inactivated?
- By adding Triton X-100
- MeM 4.6: Which commercial kit was used to measure IL-6?
- Added to the write up.
- MeM 4.7: It would be useful to specify how the Z-score and the Z-score of Ϫ were calculated.
- The obtained lab values were transformed to the z-scores using an entire population of raw scores for a given variable
Reviewer 3 Report
Comments and Suggestions for Authors
The study investigates the heterogeneity of immune responses in 106 COVID-19 patients by grouping them into four clusters based on the timing of symptom onset to hospitalization. While the progression from innate to adaptive immunity was consistent across clusters, significant differences were observed in clinical symptoms, organ damage biomarkers, and outcomes. Delayed hospitalization (Clusters C and D) was associated with increased organ damage, longer hospital stays, higher use of ECMO, and greater short-term mortality, emphasizing the critical role of organ-specific responses over immune dysregulation in determining patient outcomes. The findings challenge conventional views by highlighting the importance of organ susceptibility and damage, suggesting a need for a broader approach to managing severe viral infections.
Review Comments
- Title: The title should explicitly reference SARS-CoV-2, as the study primarily focuses on this virus. For clarity and specificity, consider changing "viral infection" to "SARS-CoV-2 infection."
- Typographical Error: There is a typo in line 93 that needs correction. Please review and revise this line for accuracy.
- Figures:
- In all figures, include the p-values to provide statistical transparency and rigor.
- Clarify how z-scores were calculated and include this explanation in the figure legends or methods section.
- Procalcitonin Variability (Line 126 and Figure 4): It appears that Cluster C demonstrated significant variability in procalcitonin levels. Ensure this finding is accurately stated and consistently represented in the text and figures.
- Figure 5: Use the full names of parameters in the figure legend and axis labels for clarity and better understanding by a broader audience.
- Figure 7 Legend and Missing Figure 8:
- The legend for Figure 7 is identical to Figure 6, which seems to be an error. Please revise it to accurately describe the content of Figure 7.
- Additionally, there is a reference to Figure 8 in line 162, but no corresponding figure is present. Clarify whether this figure was omitted or if the numbering is incorrect.
- Incomplete Sentence (Line 170): The sentence is incomplete and requires revision to convey the intended meaning clearly. Ensure all sentences are grammatically complete and contextually meaningful.
- Figure 1b Reference (Line 174): It is unclear what Figure 1b represents in this context. Review the reference to confirm its relevance and adjust the text for clarity.
- Selection of Parameters: The study examines many parameters, but it would benefit from additional explanation about the rationale behind selecting these specific markers over others. Expanding on this will enhance the scientific depth and reasoning.
- Data Collection Context:
- Provide details about when the data and patient samples were collected. This information is crucial for understanding the context and relevance of the study.
- Specify which SARS-CoV-2 variants were predominant at the time of patient recruitment, as this could significantly impact immune responses and outcomes.
Fine.
Author Response
- Title: The title should explicitly reference SARS-CoV-2, as the study primarily focuses on this virus. For clarity and specificity, consider changing "viral infection" to "SARS-CoV-2 infection."
- This work uses COVID-19 to show how infection profiles change over time. Adding this to the title may distract from the manuscript's main point. We have added sentences throughout to emphasize this. If the reviewer insists, we will reconsider changing the title.
- Typographical Error: There is a typo in line 93 that needs correction. Please review and revise this line for accuracy.
- This was revised
- In all figures, include the p-values to provide statistical transparency and rigor.
- We tried to include this information, but the figures became very crowded. We added some explanation to this matter in the Matherials and Methods section
- Clarify how z-scores were calculated and include this explanation in the figure legends or methods section.
- This was included in the materials and methods section, but we added it to the legend description.
- Procalcitonin Variability (Line 126 and Figure 4): It appears that Cluster C demonstrated significant variability in procalcitonin levels. Ensure this finding is accurately stated and consistently represented in the text and figures.
- There was an increase in positive deviation in cluster B as compared to other clusters. Legend was adjusted
- Figure 5: Use the full names of parameters in the figure legend and axis labels for clarity and better understanding by a broader audience.
- The legend was expanded.
- Figure 7 Legend and Missing Figure 8:The legend for Figure 7 is identical to Figure 6, which seems to be an error. Please revise it to accurately describe the content of Figure 7.
- Legend has been updated. Thank you for pointing this omission to us.
- Additionally, there is a reference to Figure 8 in line 162, but no corresponding figure is present. Clarify whether this figure was omitted or if the numbering is incorrect.
- This was adjusted, and the text refers to Figure 7.
- Incomplete Sentence (Line 170): The sentence is incomplete and requires revision to convey the intended meaning clearly. Ensure all sentences are grammatically complete and contextually meaningful.
- This was revised.
- Figure 1b Reference (Line 174): It is unclear what Figure 1b represents in this context. Review the reference to confirm its relevance and adjust the text for clarity.
- We expanded the legend for Figure 1
- Selection of Parameters: The study examines many parameters, but it would benefit from additional explanation about the rationale behind selecting these specific markers over others. Expanding on this will enhance the scientific depth and reasoning.
- We expanded this in the materials and methods, results, and discussion section.
- Provide details about when the data and patient samples were collected. This information is crucial for understanding the context and relevance of the study.
- We recognized this excellent remark. Appropriate information has been added in several parts of the manuscript.
- Specify which SARS-CoV-2 variants were predominant at the time of patient recruitment, as this could significantly impact immune responses and outcomes.
- This was added to the text. Recruitment took place for a few months between 3.2020 and 9.2020
Round 2
Reviewer 2 Report
Comments and Suggestions for Authors
4. Title: The title of the manuscript refers to a generic "viral infection," but the data presented only concern SARS-CoV-2-positive patients. This should be specified in the title.
· This work uses COVID-19 to show how infection profiles change over time. Adding this to the title may distract from the manuscript's main point. We have added sentences throughout to emphasize this. If the reviewer insists, we will reconsider changing the title.
The term "viral infection" is overly broad. SARS-CoV-2 cannot be considered a representative example of a generic viral infection, as it is highly likely that infections caused by entirely different viruses, such as HCV or HCMV, exhibit distinct profiles. Therefore, unless similar analyses are performed on patients with different viral infections and directly compared to those obtained from SARS-CoV-2 patients, the title should be revised accordingly.
Author Response
We understand the concern of the reviewer. We changed the title to viral illness and add one line in the discussion about generalizability of the result to other illness.. Adjusting the title, adding sentence spelling the limitation vs conveying message of the manuscript represents a balance compromise
Reviewer 3 Report
Comments and Suggestions for Authors
I have no further questions.
Author Response
Thank you for your kind words and support